# Supervised Permutation Invariant Networks for solving the CVRP with bounded fleet size

## Abstract

Learning to solve combinatorial optimization problems, such as the vehicle routing problem, offers great computational advantages over classical operations research solvers and heuristics. The recently developed deep reinforcement learning approaches either improve an initially given solution iteratively or sequentially construct a set of individual tours. However, most of the existing learning-based approaches are not able to work for a fixed number of vehicles and thus bypass the complex assignment problem of the customers onto an apriori given number of available vehicles. On the other hand, this makes them less suitable for real applications, as many logistic service providers rely on solutions provided for a specific bounded fleet size and cannot accommodate short term changes to the number of vehicles. In contrast we propose a powerful supervised deep learning framework that constructs a complete tour plan from scratch while respecting an apriori fixed number of available vehicles. In combination with an efficient post-processing scheme, our supervised approach is not only much faster and easier to train but also achieves competitive results that incorporate the practical aspect of vehicle costs. In thorough controlled experiments we compare our method to multiple state-of-the-art approaches where we demonstrate stable performance, while utilizing less vehicles and shed some light on existent inconsistencies in the experimentation protocols of the related work.

## 1 Introduction

The recent progress in deep learning approaches for solving combinatorial optimization problems (COPs) has shown that specific subclasses of these problems can now be solved efficiently through training a parameterized model on a distribution of problem instances. This has most prominently been demonstrated for the vehicle routing problem (VRP) and its variants (Kool et al. (2019); Chen & Tian (2019); Xin et al. (2021)). The current leading approaches model the VRP either as a local search problem, where an initial solution is iteratively improved or as a sequential construction process successively adding customer nodes to individual tours until a solution is achieved. Both types of approaches bypass the implicit bin-packing problem that assigns packages (the customers and their demands) to a pre-defined maximum number of bins (the vehicles). We show that this assignment can be learned explicitly while also minimizing the main component of the vehicle routing objective, i.e. the total tour length. Furthermore, finding a tour plan for a fixed fleet size constitutes an essential requirement in many practical applications. To this end, many small or medium-sized service providers cannot accommodate fleet size adjustments where they require additional drivers on short notice or where very high acquisition costs prohibit the dynamic extension of the fleet. Figure 1 shows the variation of fleet sizes for different problem sizes. For all problem sizes the baseline approaches (green, red, purple) employ more vehicles than our approach (blue, orange) and thereby inquire potentially more costs by requiring more vehicles. In contrast our vanilla approach (blue) guarantees to solve the respective problems with exactly the apriori available number of vehicles (4, 7, 11 for the VRP20, VRP50, VRP100 respectively) or less. Our approach of learning to construct a complete, valid tour plan that assigns all vehicles a set of customers at once has so far not been applied to the VRP. We amend and extend the Permutation Invariant Pooling Model (Kaempfer & Wolf (2018)) that has been applied solely to the simpler multiple Traveling Salesmen Problem (mTSP), for which also multiple tours need to be constructed, but are albeit not subject to any capacity constraints.

In this work we propose an end-to-end learning framework that takes customer-coordinates, their associated demands and the maximal number of available vehicles as inputs to output a bounded number of complete tours and show that the model not only learns how to optimally conjunct customers and adhere to capacity constraints, but that it also outperforms the learning-based reinforcement learning (RL) baselines when taking into account the total routing costs. By adopting a full-fledged supervised learning strategy, we contribute the first framework of this kind for the capacitated VRP that is not only faster and less cumbersome to train, but also demonstrates that supervised methods are able, in particular under practical circumstances, to outperform state-of-the-art RL models.

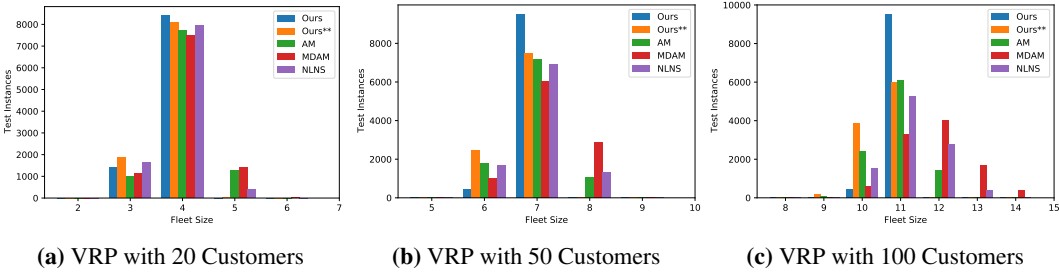

**(a)** VRP with 20 Customers    **(b)** VRP with 50 Customers    **(c)** VRP with 100 Customers

**Figure 1:** Histograms over the fleet sizes used for solving the VRP. Our approach (blue) is able to solve the VRP with 20, 50, 100 customers with 4, 7 and 11 vehicles respectively, while the baseline approaches (see section 5) utilize considerably more vehicles. Our approach (orange) refers to the setting of where we ensure a solution to be found (see section 4.3)

The contributions of this work can be summarized as follows:

- *Tour-plan construction approach for the capacitated VRP with fixed Vehicle Costs*: We propose a deep learning framework that *learns to solve* the NP-hard capacitated VRP and constructs a complete tour-plan for an specified fleet size. In this way it ensures to solve the problem for an apriori fixed number of vehicles.

- *Supervised learning for discrete optimization*: The proposed approach is the first ML-based construction approach for the VRP that relies *entirely* on supervised learning to output feasible tours for all available vehicles.

- *Competitiveness*: We compare our model against prominent state-of-the-art deep learning and OR solvers through a thorough and unified experimental evaluation protocol that ensures comparability amongst different ML-based approaches. We show that our approach delivers competitive results in comparison to approaches that work only for unbounded fleet sizes and outperforms these methods when considering fixed vehicle costs.

## 2 RELATED WORK

**Learned Construction Heuristics.** The work by Vinyals et al. (2015) introducing *Pointer Networks (PtrNet)* revived the idea of developing end-to-end deep learning methods to construct solutions for COPs, particularly for the TSP, by leveraging an encoder-decoder architecture trained in a supervised fashion. Bello et al. (2016) and Khalil et al. (2017) proposed RL to train the Pointer Network for the TSP, while Nazari et al. (2018) extended the PtrNet in order to make the model invariant to the order of the input sequence and applied it to the CVRP. The *Attention Model (AM)* described in Kool et al. (2019) features an attention based encoder-decoder model that follows the Transformer architecture (Vaswani et al. (2017)) to output solution sequences for the TSP and two variants of the CVRP. Xin et al. (2021) extend the AM by replacing the single decoder with multiple, identical decoders with no shared parameters to improve diversity among the generated solutions. Besides these sequential approaches, "heatmap" construction approaches that learn to assign probabilities to given edges in a graph representation through supervised learning have recently been applied to routing problems. Joshi et al. (2019) use a Graph Convolutional Network to construct TSP tours and show that by utilizing a parallelized beam search, auto-regressive construction approaches for the TSP can be outperformed. Kool et al. (2021) extend the proposed model by Joshi et al. (2019) for the CVRP while creating a hybrid approach that initiates partial solutions using a heatmap representation as a preprocessing step, before training a policy to create partial solutions and refining these through

dynamic programming. Kaempfer & Wolf (2018) extend the learned heatmap approach to the number of tours to be constructed; their *Permutation Invariant Pooling Network* addresses the mTSP (a TSP involving multiple tours but no additional capacity constraints), where feasible solutions are obtained via a beam search and have been proven to outperform a meta-heuristic mTSP solver.

**Learned Search Heuristics.** In contrast to construction heuristics that build solutions to routing problems from scratch, learned search heuristics utilize well-known local search frameworks commonly adopted in the field of operations research (OR) to learn to improve an initial given solution in an iterative fashion through RL. The *NeuRewriter* model (Chen & Tian (2019)) learns a policy that is composed of a "rule-picking" and a "region-picking" part in order to iteratively refine a VRP solution and demonstrates superior performance over certain auto-regressive approaches (Nazari et al. (2018); Kool et al. (2019)). Similarly, the *Learning to Improve* approach by Lu et al. (2020) learns a policy that iteratively selects a local search operator to apply to the current solution and delivers new state-of-the-art results amongst existing machine learning heuristics. However, it is computationally infeasible during inference in some cases taking more than thirty minutes to solve a single instance, which makes this approach incomparable to most other methods. In contrast, *Neural Large Neighborhood Search (NLNS)* (Hottung & Tierney (2020)) is based on an attention mechanism and integrates learned OR-typical operators into the search procedure, demonstrating a fair balance between solution quality and computational efficiency, but can only be evaluated in batches of thousand instances in order to provide competitive results.

## 3 PROBLEM FORMULATION

This section introduces the capacitated vehicle routing problem which implicitly encompasses a bin packing problem through finding a feasible assignment of customer nodes to a given set of vehicles. We showcase how existing ML-based approaches circumvent this property and solve a simpler variant of the problem. We then demonstrate how to cast the VRP into a supervised machine learning task and finally propose an evaluation metric that respects the utilization of vehicles in terms of fixed vehicle costs.

### 3.1 THE CAPACITATED VEHICLE ROUTING PROBLEM

Following Baldacci et al. (2007) the Capacitated Vehicle Routing Problem (CVRP) can be defined in terms of a graph theoretical problem. It involves a graph $G = (V, E)$ with vertex set $V = \{0, ..., N\}$ and edge set $E$. Each customer $i \in V_c = \{1, ..., N\}$ has a demand $q_i$ which needs to be satisfied and each weighted edge $\{i, j\} \in E$ represents a non-negative travel cost $d_{ij}$. The vertex $0 \in V$ represents the depot. A set $M$ of identical (homogeneous) vehicles with same capacity $Q$ is available at the depot. The general CVRP formulation sets forth that "all *available vehicles* must be used" (Baldacci et al. (2007, p. 271)) and $M$ cannot be smaller than $M_{\min}$, corresponding to the minimal fleet size needed to solve the problem. Accordingly, the CVRP consists of finding *exactly* $M$ simple cycles with minimal total cost corresponding to the sum of edges belonging to the tours substitute to the following constraints: (i) each tour starts and ends at the depot, (ii) each customer is served exactly once and (iii) the sum of customer demands on each tour cannot be larger than $Q$. A formal mathematical definition of the problem can be found in Appendix A.

Therefore the solution requires exactly $M$ cycles to be found, which requires in turn the value of $M$ to be set in an apriori fashion. However, already the task of finding a *feasible* solution with exactly $M$ tours is NP-complete, thus many methods choose to work with an unbounded fleet size (Cordeau et al. (2007, p. 375)) which guarantees a feasible solution by simply increasing $M$ if there are any unvisited customers left. In contrast, our method tackles the difficulty of the assignment problem by jointly learning the optimal sequence of customer nodes and the corresponding optimal allocation of customers to a fixed number of vehicles. In order to do so, we rely on generated near-optimal targets that determine $M_{\min}$ for each problem instance in the training set.

### 3.2 THE VRP AS A SUPERVISED MACHINE LEARNING PROBLEM

The goal of the supervised problem is to find an assignment $\hat{\mathbf{Y}}$ that allocates optimally ordered sequences of customers to vehicles. Respectively each VRP instance $X$ which defines a single graph theoretical problem, is characterised by the following *three entities*:

- A set of $N$ customers. Each customer $i$, $i \in \{1, ..., N\}$ is represented by the features $\mathbf{x}_i^{\text{cus}}$.

- $M$ vehicles, where each vehicle $k \in \{1, ..., M\}$ is characterised by the features $\mathbf{x}_k^{\text{veh}}$.

- The depot, the 0-index of the $N$ customers, is represented by a feature vector $\mathbf{x}^{\text{dep}}$.

Consequently, a single VRP instance is represented as the set $X = (\mathbf{x}^{\text{dep}}; \mathbf{x}_{1,...,n}^{\text{cus}}; \mathbf{x}_{1,...M}^{\text{veh}})$. The corresponding ground truth target reflects the near-optimal tour plan and is represented by a binary tensor $\mathbf{Y}^{M \times N \times N}$, where $\mathbf{Y}_{k,i,j} = 1$ if the $k^{\text{th}}$ vehicle travels from customer node $i$ to customer node $j$ in the solution and else $\mathbf{Y}_{k,i,j} = 0$.

Let $\mathcal{X}$ represent the predictor space populated by VRP instances $X$ and let $\mathcal{Y}$ define the target space comprising a population of target instances $\mathbf{Y}$. The proposed model is trained to solve the following assignment problem: Given a sample $\mathcal{D} \in (\mathcal{X} \times \mathcal{Y})^*$ from an unknown distribution $p$ and a loss function $\ell : \mathcal{Y} \times \mathcal{Y} \rightarrow \mathbb{R}$, find a model $\hat{y} : \mathcal{X} \rightarrow \mathcal{Y}$ which minimizes the expected loss, i.e.

$$\min \ \mathbb{E}_{(x,y) \sim p} \ \ell(y, \hat{y}(x)) \tag{1}$$

For a sampled predictor set $x$, the model $\hat{y}(x)$ outputs a stochastic adjacency tensor $\hat{\mathbf{Y}}^{M \times N \times N}$ that consists of the approximate probability distribution defining a tour plan.

## 3.3 Fixed Vehicle Costs

In order to highlight the importance of the assignment problem incorporated in the CVRP and the possible impact of an unlimited fleet sizes in real world use cases, we introduce fixed vehicle costs in the evaluation. As mentioned above, a problem instance is theoretically always "solvable" in terms of feasibilty by any method that operates on the basis of unbounded fleet sizes. Nevertheless, the increase in the number of tours is undisputedly also a cost driver besides the mere minimization of route lengths. Thus, we compare our approach to state-of-the-art RL approaches that do not limit the number of tours in a more holistic and realistic setting via the metric that we denote $\text{Cost}_v$:

$$\text{Cost}_v = \sum_{k,i,j} d_{ij} \mathbf{Y}_{k,i,j} + c_v \sum_k \mathbb{1}(\mathbf{Y}_{k,0,0} = 0) \tag{2}$$

where $c_v$ is the fixed cost per vehicle *used*, such that if a vehicle $k$ leaves the depot, we have $\mathbf{Y}_{k,0,0} = 0$, else we indicate that this vehicle remains at the depot with $\mathbf{Y}_{k,0,0} = 1$. Concerning the value of $c_v$ we rely on data used in the *Fleet Size and Mix VRP* in Golden et al. (1984) and find matching values corresponding to the capacities used in our experimental setting. To this end, we incorporate fixed costs of $c_v = 35$, $c_v = 50$ and $c_v = 100$ for the VRP with 20, 50 and 100 customers respectively (a summarizing Table can be found in Appendix C.2). We want to note that even though the RL baselines are not specifically trained to minimize this more realistic cost function, our model is neither, as will be shown in section 4. This realistic cost setting functions as mere statistical evaluation metric to showcase the implications of fixed vehicle costs.

## 4 Permutation Invariant VRP Model

The proposed model outputs a complete feasible tour-plan for the CVRP in the form of a stochastic adjacency tensor $\hat{\mathbf{Y}}$. Inspired by the Transformer-based architecture in Kaempfer & Wolf (2018) that tackles the mTSP, we extend this architecture to additionally capture vehicle capacity constraints. The framework encompasses three components: (i) an embedding layer (*Encoder*), (ii) an information extraction mechanism (*Pooling*) and (iii) an output construction block (*Decoder*). Figure 2 displays an overview of the model architecture.

## 4.1 MODEL ARCHITECTURE

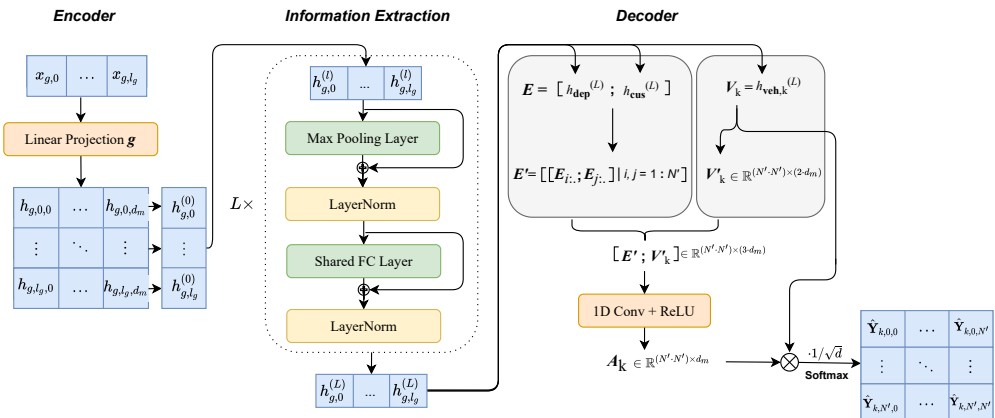

**Figure 2:** Permutation Invariant VRP model. Framework overview.

*Embedding.* The three entities that make up a VRP instance $X$ are embedded separately, such that each encoding is embedded into a shared dimensional space $d_m$ by a linear projection:

$$h_g^{(0)} = W_g \mathbf{x}^g + b_g \quad \text{for g} \in \{\text{dep, cus, veh}\} \tag{3}$$

where $h_g^{(0)} \in \mathbb{R}^{l_g \times d_m}$ is the initial embedding of each entity that will remain of different lengths $l_{\text{cus}} = N, l_{\text{veh}} = M, l_{\text{dep}} = 1$. The separate embedding for each entity and the architecture of the network ensure that the order of the elements in the entities is not relevant for the operations performed in the network, which establishes the permutation invariance.

*Information Extraction.* The information extraction component iteratively pools contexts across entities and linearly transforms these context vectors $c_{g,1}, \ldots, c_{g,l_g}$ with the entity embeddings of the current layer $h_g^l$. For each element $r \in \{1, \ldots, l_g\}$ in an entity the contexts are pooled:

$$c_{g,r}^{(l)} = \text{pool}(h_{g,1}^{(l)}, \ldots, h_{g,l_g}^{(l)}) \quad \text{for g} \in \{\text{dep, cus, veh}\} \tag{4}$$

For the next layer's entity representation, the contexts and previous embeddings are linearly projected:

$$h_g^{(l+1)} = f_g([h_g^{(l)}; c_g^{(l)}]) \quad \text{for g} \in \{\text{dep, cus, veh}\} \tag{5}$$

This pooling mechanism proceeds until a final representation of each entity is retrieved; $h_{\text{dep}}^{(L)}$, $h_{\text{cus}}^{(L)}$ and $h_{\text{veh}}^{(L)}$. These are then passed to the decoding procedure. A more detailed description of the operations performed in the Pooling Layer is described in Appendix C.3.

*Decoder.* The decoding step in the model constructs the output tensor $\hat{\mathbf{Y}} \in \mathbb{R}^{M \times N' \times N'}$ where $N' = N + 1$ indicates the size of the full adjacency tensor including the depot. A preliminary step consists of forming a feature representation of all edges (potential paths) between all pairs of vertices in the graph, denoted as $E'$:

$$
\begin{aligned}
E &= [h_{\text{dep}}^{(L)}; h_{\text{cus},1}^{(L)}; \ldots; h_{\text{cus},N}^{(L)}] &&\in \mathbb{R}^{N' \times d_m} \\
E' &= [[E_{i:.}; E_{j:.}] \mid i, j = 1 : N'] &&\in \mathbb{R}^{(N' \cdot N') \times 2 \cdot d_m}
\end{aligned}
\tag{6}
$$

The final construction procedure combines the edge ($E'$) and fleet ($V = h_{\text{veh}}^{(L)}$) representations where each vehicle representation $V_k = h_{\text{veh},k}^{(L)}$, $k \in M$ enters the combination process twice:

1. In the linear transformation with the edges $E'$, where $W_o$ and bias $b_o$ are the learned weights and $V_k'$ is an expanded version of $V_k$ to match the dimensionality of $E'$:

$$A_k = \text{ReLU}(W_o[E'; V_k'] + b_o) \tag{7}$$

2. In a scaled dot-product which returns compatibility scores for each path potentially travelled by vehicle $k$ to emphasize a direct interaction between vehicles and convolved edges:

$$\hat{\mathbf{Y}}_k = \text{softmax}(\frac{A_k^T V_k'}{\sqrt{d_m}}) \tag{8}$$

Finally a softmax is applied to transform the compatibility scores of vehicles and edges into probabilities $\hat{\mathbf{Y}} \in \mathbb{R}^{M \times N' \times N'}$.

## 4.2 SOLUTION DECODING

In order to transform the doubly-stochastic probability tensor $\hat{\mathbf{Y}}$ into discrete tours in terms of a binary assignment, we use a greedy decoding strategy. In *training*, a pseudo-greedy decoding (see Appendix C.4) renders potentially infeasible solutions by transforming $\hat{\mathbf{Y}}$ into a binary assignment tensor $\hat{\mathbf{Y}}^*$, where $\hat{\mathbf{Y}}^* = 1$ for the *predicted* path of vehicle $k$ between vertices $i$ and $j$ and 0 otherwise.

---

**Input:** $\hat{\mathbf{Y}} \in \mathbb{R}^{M \times N' \times N'}, \hat{\mathbf{Y}}^* \in \{0,1\}^{M \times N' \times N'}, q \in \mathbb{R}^{N'}, Q' \in \mathbb{R}^M, U, Q \in \mathbb{R}$
**Output:** Routes $\hat{\mathbf{Y}}^* \in \{0,1\}^{M \times N' \times N'}$
1: **for** $i \in U$ **do**
2:     **if** $Q' - q_i < 0_M$ **then**
3:         $M \leftarrow M + 1$ ;              ▷ Add new tour if "guarantee solution" is set
4:         $\hat{\mathbf{Y}}^*_{M,\cdot,\cdot} \leftarrow 0$ ;                  ▷ Update binary solution tensor
5:         $Q'_M \leftarrow Q$;                     ▷ Update capacity tensor
6:     **end if**
7:     $v \leftarrow \arg\max(Q' - q_i)$ ;          ▷ Vehicle with max capacity left
8:     $V_a \leftarrow \text{argselect}(\hat{\mathbf{Y}}^*_v > 0)$ ;    ▷ Select all possible vertices for insertion
9:     $j_{\text{before}} \leftarrow V_a[\arg\max(\hat{\mathbf{Y}}_{v,V_a,i})]$ ;  ▷ Assign incoming edge of inserted customer
10:    $j_{\text{after}} \leftarrow \arg\max(\hat{\mathbf{Y}}^*_{v,j_{\text{before}},:})$ ;    ▷ Assign outgoing edge of inserted customer
11:    $\hat{\mathbf{Y}}^*_{v,j_{\text{before}},:} \leftarrow 0.0$ ;              ▷ Update Solution Tensor
12:    $\hat{\mathbf{Y}}^*_{v,j_{\text{before}},i} \leftarrow 1.0$
13:    $\hat{\mathbf{Y}}^*_{v,i,j_{\text{before}}} \leftarrow 1.0$
14:    $Q'_v \leftarrow Q'_v - q_i$ ;             ▷ Update $v$'s capacity after insertion
15: **end for**
16: **return** $\hat{\mathbf{Y}}^*$

**Algorithm 1:** Repair Greedy Solution. $q$ is the customer's demand vector, $Q'$ is the vector remaining capacity for all vehicles.

---

During *inference*, the pseudo-greedy decoding is substituted by a strictly greedy decoding, and thereby accepts only tour-plans, which do not violate the respective capacity constraint. This is done by tracking the remaining capacity of all vehicles and masking nodes that would surpass the capacity. Any un-assigned customers violating capacity constraints in the final state of the algorithm are recovered in a list $U$ and passed as input to a repair procedure for $\hat{\mathbf{Y}}^*$ (Algorithm 1). For each unassigned customer in list $U$, Algorithm 1 assigns customer $i$ to the vehicle with most remaining capacity and positions the missing customer between $j_{\text{before}}$ and $j_{\text{after}}$ according to the distribution $\hat{\mathbf{Y}}$, before updating the predicted binary solution tensor $\hat{\mathbf{Y}}^*$ and the remaining capacity $Q'$ accordingly.

## 4.3 INFERENCE SOLUTION POST-PROCESSING

Unfortunately the repair operation of Algorithm 1 leads to a situation where the resulting tours during inference are not necessarily local optima anymore. To improve these tours we add a heuristic post processing procedure doing several iterations of local search via the google OR-tools solver (Perron & Furnon (2019)). Given a valid solution for a particular number of vehicles, the solver runs a (potentially time-limited) search that improves the initial solution $\hat{\mathbf{Y}}^*$. During inference, the method optionally can relax the bound on the fleet size to guarantee to find a solution which is important in cases where instances are particularly difficult to solve due to a very tight bound on capacity.

This is done by initially adding an artificial tour to the plan for the remaining missing customers before running the post-processing scheme. Thus, this mechanism enables our method to also solve instances it is not initially trained for to provide maximum flexibility.

## 4.4 TRAINING THE PERMUTATION INVARIANT VRP MODEL

In order to train our model to learn the assignment of customer nodes to the available vehicles and adhere to capacity constraints, we extend the original negative log-likelihood loss by a penalty formulation ($L_{\text{over}}$) and a auxiliary load loss, controlled via weights $\alpha_{\text{over}}$ and $\alpha_{\text{load}}$.

The model's characteristic of permutation invariance induces the necessity for the training loss to be agnostic to the vehicle-order as well as the travel direction of the tours. Concerning the route-direction, we denote $\mathbf{Y}_{k,i,j}^0 = \mathbf{Y}_{k,i,j}$ and $\mathbf{Y}_{k,i,j}^1 = \mathbf{Y}_{k,j,i}$ Therefore, the loss to be minimized is the minimum (penalized) normalized negative log-likelihood of $\hat{\mathbf{Y}}$ with respect to the sampled target $\mathbf{Y}$:

$$\mathcal{L}(\hat{\mathbf{Y}}) := \min_{\pi, b \in \{0,1\}^M} -\sum_{k=1}^M \left( \sum_{i,j=0}^N \mathbf{Y}_{\pi(k),i,j}^{b_k} \log \hat{\mathbf{Y}}_{k,i,j} \right) + \alpha_{\text{load}} |\text{load}(\hat{Y}_{k,.,.}) - \text{load}(Y_{\pi(k),.,.})| \tag{9}$$
$$+ \alpha_{\text{over}} L_{\text{over}}(\text{load}(\hat{Y}_{k,.,.}))$$

with the load of a tour $T = \mathbf{Y}_{k,.,.} \in \{0,1\}^{N' \times N'}$

$$\text{load}(T) := \sum_{i,j=0}^{N'} T_{i,j} q_i \tag{10}$$

and a shifted quadratic error for overloading

$$L_{\text{over}}(q) := \begin{cases} 0, & \text{if } q \le Q \\ (1 + q - Q)^2, & \text{else} \end{cases} \tag{11}$$

The calculation of the loss formulation in Equation 9 would induce considerable computational overhead, therefore a less memory-intensive formulation of the same loss is implemented (see Appendix C.5). The extensions to the original Permutation Invariant Pooling Network are summarized in Appendix C.6.

## 5 EXPERIMENTS

In the following experiments we want to showcase and validate our main contributions:

1. Our method works for problems with bounded fleet sizes and outperforms state-of-the-art RL models when accounting for fixed vehicle costs.

2. Our supervised learning model is fast to train and delivers competitive results in comparison to the state-of-the-art, while utilizing considerably less vehicles.

We consider three different problem sizes, with 20, 50 and 100 customer vertices respectively and generate solvable training instances following a rejection-sampled version of the data generating process in Kool et al. (2019). The near-optimal target instances are generated with google's OR-Tools guided local search solver (Perron & Furnon (2019)), where we set $M = 4, 7, 11$ and $Q = 30, 40, 50$ for each problem size respectively. In total we generate roughly 100,000 training samples per problem size. The hyperparameter settings are described in Appendix D.1. For the evaluation, we work with two versions of the test dataset in Kool et al. (2019), the originally provided one and a *rejection sampled* version of it, as technically our model is trained to solve only the instances for which the lower bound $\sum_{i \in n} q_i \le MQ$ holds. We compare our model to recent RL methods that comprise leading autoregressive (AM (Kool et al. (2019)), MDAM (Xin et al. (2021))) as well as a search based (NLNS (Hottung & Tierney (2020))) approaches. The NeurRewriter (Chen & Tian (2019)) and L2I model (Lu et al. (2020)) are not re-evaluated due to extensive training- and inference times. Details

concerning the re-evaluated (and not re-evaluated) baselines are found in Appendix D.2. Furthermore, we revisit different published results in a *per-instance* re-evaluation to elevate comparability and demonstrate strengths and weaknesses of the methods. We note that most methods were originally evaluated in a per-batch fashion where they can leverage the massive parallelization capabilities of current graphical processing units, while that setting is of less practical relevance.

## 5.1 FIXED VEHICLE COSTS

To validate the first point of our main contribution, we evaluate the performance of different RL methods and our own model on the metric defined in Equation 2. Table 1 shows the results for evaluating the methods on the total tour length (Cost), as well as the measure incorporating fixed costs for each vehicle ($Cost_v$). The results in Table 1 shows that method outperforms state-of-the-art

**Table 1:** Results for different learning-based models on 10000 **rejection sampled** VRP Test Instances. Cost including fixed vehicle costs ($Cost_v$) and without (Cost) is reported. Regarding the percentage of solved instances we achieve 99%, 99.5% and 100% coverage for the problem sizes 20, 50 and 100 respectively.

| Model | VRP20 | | | VRP50 | | | VRP100 | | |
|---|---|---|---|---|---|---|---|---|---|
| | Cost | $Cost_v$ | $t$/inst | Cost | $Cost_v$ | $t$/inst | Cost | $Cost_v$ | $t$/inst |
| NLNS | 6.22 | 145.0 | 1.21s | 10.95 | 367.3 | 2.01s | 17.18 | 913.3 | 3.02s |
| AM (greedy) | 6.33 | 147.4 | 0.04s | 10.90 | 357.0 | 0.12s | 16.68 | 888.4 | 0.22s |
| AM (sampl.) | 6.18 | 143.7 | 0.05s | 10.54 | 352.2 | 0.19s | 16.12 | 873.4 | 0.57s |
| MDAM (greedy) | 6.21 | 147.3 | 0.46s | 10.80 | 370.40 | 1.08s | 16.81 | 961.3 | 2.02s |
| MDAM (bm30) | 6.17 | 140.3 | 5.06s | 10.46 | 351.66 | 9.00s | 16.18 | 889.1 | 16.58s |
| Ours | **6.16** | **135.5** | 0.05s | 10.76 | **346.1** | 0.16s | 16.93 | **859.6** | 0.84s |
| Ours** | **6.17** | **136.0** | 0.05s | 10.77 | **346.1** | 0.16s | 16.93 | **859.6** | 0.84s |

(**) With the option of a guaranteed solution

RL methods on the VRP with fixed vehicle costs across all sizes of the problem and even outperforms them on the plain total tour length minimization cost for the VRP with 20 customers, while delivering the shortest inference times per instance together with the AM Model. Furthermore, our method's results do not greatly differ whether we set the option of guaranteeing a solution (last row) to True versus accepting that there are unsolved samples, speaking for the strength of our model for being able to solve both problems incorporated in the VRP at once, the feasible assignment problem to a particular number vehicles and the minimization of the total tour length. This is also represented in Figure 1; while the amount of solutions solved with more than the apriori fixed fleet size is vanishingly small for the VRP20 and VRP50 it is exactly 0 for the VRP100. For the VRP of size 50 the RL-based methods only outperform our greedily evaluated method on the routing length minimization when employing sampling or a beam search during inference, while being worse when used greedily. Concerning the problem with 100 customers, our method is outperformed by the RL approaches on the vanilla total route length cost, but solves the majority of the problems with a significantly smaller fleet size, reflected by the considerably smaller total costs ($Cost_v$). This is strongly supported by Figure 1, where we see that especially for the VRP with 100 customers, the RL-based methods consistently require more vehicles than needed.

## 5.2 COMPARATIVE RESULTS ON THE BENCHMARK DATASET

We want to assess the competitiveness of our method also on the literature's bench-marking test set provided in Kool et al. (2019). Technically, these instances are not all solvable by our model, as it is not trained to solve problems where $\sum_{i \in n} q_i \leq MQ$. In Table 2 we therefore indicate the percentage of solved instances by our plain model in brackets. Looking at Table 2, our method's performance is generally on par with or slightly better than RL-based construction methods when employing a greedy decoding. Concerning the per-instance run times, MDAM and the NLNS take orders of magnitude longer than the AM model and our method. We acknowledge that exploiting parallelism to enhance computational efficiency is a justified methodological choice, but in terms of comparability, we argue for the pragmatic approach of comparing per-instance runtimes to remedy problematic comparison of models using different architectures and mini batch sizes. For completeness Table 11 in Appendix D.3 illustrates the per-batch evaluation of the baselines. Appendix D.2 also discusses

the discrepancies in performances of NLNS and MDAM with respect to their published results. Even though the solution quality and competitiveness of our method decrease when considering larger problem sizes, the per-instance inference time remains among the best. Nevertheless, we want to emphasize that the baselines in general require more tours than our method potentially leading to higher costs when employed in real-world scenarios.

**Table 2:** Results for different ML and OR Models on 10000 VRP Test Instances (Kool et al. (2019)).

| Model | VRP20 | | VRP50 | | VRP100 | |
|---|---|---|---|---|---|---|
| | Cost | $t$ | Cost | $t$ | Cost | $t$ |
| Gurobi | 6.10 | - | - | - | - | |
| LKH3 | 6.14 | 2h | 10.38 | 7h | 15.65 | 13h |
| | Cost | $t$/inst | Cost | $t$/inst | Cost | $t$/inst |
| NLNS (t-limit) | 6.24 | 2.41s | 10.96 | 2.41s | 17.72 | 2.02s |
| AM (greedy) | 6.40 | 0.04s | 10.98 | 0.09s | 16.80 | 0.17s |
| AM (sampl.) | 6.25 | 0.05s | 10.62 | 0.17s | 16.20 | 0.51s |
| MDAM (greedy) | 6.28 | 0.46s | 10.88 | 1.10s | 16.89 | 2.00s |
| MDAM (bm30) | 6.15 | 5.06s | 10.54 | 9.00s | 16.26 | 16.56s |
| Ours | 6.18 (94%*) | 0.05s | 10.81 (94%*) | 0.16s | 16.98 (98%*) | 0.82s |
| Ours** | 6.24 | 0.05s | 10.87 | 0.17s | 17.02 | 0.82s |

(*) Percentage of solved instances. (**) With the option of a guaranteed solution

## 5.3 TRAINING TIMES

We showcase the fast training of our model against the NeuRewriter model (Chen & Tian (2019)) as a representative for RL methods. Table 3 shows the results from runs on an A100-SXM4 machine with 40GB GPU RAM. To train the NeuRewriter for the VRP50 and VRP100 with the proposed batch size of 64, 40GB GPU RAM are not sufficient. Instead, we used a batch size of 32 and 16 respectively, where the total runtime is estimated for completing 10 epochs. Accounting for our dataset generation, we note that with 2 Xeon Gold 6230 CPU cores, it takes roughly 11 days to generate one dataset. That said, we emphasize that the OR Tools is straightforwardly parallelizable (emberrassingly parallel!), such that doubling the amount of cores, roughly halves the runtime.

**Table 3:** Training Times per Epoch and Total Train Time of NeuRewriter (Chen & Tian (2019)) and our Model

| | VRP20 | | VRP50 | | VRP100 | |
|---|---|---|---|---|---|---|
| | $t$/epoch | $t$ | $t$/epoch | $t$ | $t$/epoch | $t$ |
| NeuRewriter | 15h | 6d | > 31h | > 13d | > 36h | > 15d |
| Ours | 4m | 4h | 5.4m | 4.5h | 1h | 1.5d |

## 6 CONCLUSION AND FUTURE WORK

With the proposed supervised approach for solving the VRP, we investigate a new road to tackle routing problems with machine learning that comes with the benefit of being fast to train and which is able to produce feasible solutions for a apriori fixed fleet size. We show that our model is able to jointly learn to solve the VRP assignment problem and the route length minimization. Our work focuses on and showcases practical aspects for solving the VRP that are important for decision makers in the planning industry and shows that our method outperforms existing models when accounting for fixed vehicle costs. In future work, we aim to alleviate the computational shortcomings of the train loss calculation, such that the model's fast training capability can be extended to problems with larger fleet sizes.

## 7 REPRODUCIBILITY STATEMENT

Considering the current pace at which new approaches and methods are emerging, not only but also for the research area of learning based combinatorial optimization, fast and easy access to source codes with rigorous documentation needs to be established in order to ensure comparability and verified state-of-the-arts. To this end, we will provide the code for our fast supervised learning method on a github repository, which will be publicly available. During the review process, the code will be made available to he reviewers. On a broader scope, since the findings in this paper comprise not only our own results, but also those of re-evaluated existing approaches in the field, we plan to develop a benchmark suite on learning-based routing methods for which the re-evaluations and produced codes will build the basis for.

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

## A  FORMAL DEFINITIONS

### A.1  MATHEMATICAL FORMULATION OF THE CVRP

Following the notation in section 3.1 of the paper, the CVRP can be formalized as the following two-index vehicle flow formulation, first introduced by Laporte et al. (1985):

Given a binary decision variable $x_{ij}$ taking either a value of 1 or 0  $\forall \{i,j\} \in E \; \{\{0,j\} : j \in V_c\}$ where $x_{ij} = 1$ if the path $\{i,j\}$ is traversed and $x_{0j} = 2$ if a tour including the customer $j$ is selected in the solution, solve the following integer program:

$$\min \quad \sum_{\{i,j\} \in E} d_{ij} x_{ij} \tag{12a}$$

$$s.t. \sum_{\{i,j\} \in \delta(\{h\})} x_{ij} = 2 \quad (\forall h \in V_c) \tag{12b}$$

$$\sum_{\{i,j\} \in \delta(S)} x_{ij} \geq 2[q(S)/Q] \quad (\forall S \in \mathscr{S}) \tag{12c}$$

$$\sum_{j \in V_c} x_{0j} = 2M \tag{12d}$$

$$x_{ij} \in \{0,1\} \quad (\forall \{i,j\} \in E \; \{\{0,j\} : j \in V_c\}) \tag{12e}$$

$$x_{0j} \in \{0,1,2\} \quad (\forall \{0,j\}, j \in V_c\}) \tag{12f}$$

where a subset of customers is denoted by $S \subseteq V_c$,  $\delta(S) = \{i,j\} \in E : i \in S, j \notin S$ or $i \notin S, j \in S$ is the cutset defined by $S$ and $\mathscr{S} = \{S : S \subseteq V_c, |S| \geq 2\}$.

Equations (12b) lists the degree constraints. Equation (12c) indicates that for any subset $S$ of customers, excluding the depot, at least $[q(S)/Q]$ vehicles enter and leave it. Constraint (12d) ensures that $M$ vehicles leave and return to the depot. Equations (12e) and (12f) constitute the integrality constraints.

### A.2  BIN PACKING PROBLEM

The Bin Packing Problem can be formulated in words as follows:

Given a set of $n$ items $\{1, \ldots, n\}$, where $w_j$ is the weight of the item $j \in \{1, \ldots, n\}$ and $n$ bins, each with capacity $C$. Assign each item to one bin so that:

- The total weight of items in each bin does not exceed $C$

- The total number of utilized bins is minimized.

In mathematical terms, this results in solving the following minimization problem, where $x_{ij} \in \{0, 1\}$ is equal to 1 if item $j$ is assigned to bin $i$, and $z_i \in \{0, 1\}$ is equal to 1 if bin $i$ is used:

$$\min \quad \sum_i^n z_i \tag{13a}$$

$$s.t. \quad \sum_j^n w_j x_{ij} \leq C z_i, \quad i = 1, \ldots, n \tag{13b}$$

$$\sum_i^n x_{ij} \quad = 1, \quad j = 1, \ldots, n \tag{13c}$$

$$x_{ij} \in \{0, 1\} \quad i = 1, \ldots, n \tag{13d}$$

$$z_i \in \{0, 1\} \quad i = 1, \ldots, n \tag{13e}$$

## B  ABLATIONS

This section demonstrates that the changes in our approach and notably to the loss structure of the approach, compared to the original model in Kaempfer & Wolf (2018), are improving the model's performance and efficiency. The results summarized in Table 2 and 3 are retrieved form an evaluation on the rejection sampled Kool et al. Dataset with the option of a guaranteed solution.

### B.1  ADDITIONAL LOSS STRUCTURES

This section demonstrates that the additional loss structures (Penalty and auxiliary Load Loss) imposed on the pure assignment Loss help reduce the capacity violation during training and guide the model towards a better performance. The right hand side of Figure 3 shows that with the two additional loss structures, the model achieves an average capacity violation of 0.017 per vehicle, while the model, when trained with the plain assignment loss, still exhibit an average capacity violation of 0.021 and even starts increasing again after 40 epochs. While an average capacity violation of 0.017 per vehicle, still means that the model cannot control the capacity constraint to a hundred percent, this still significantly helps the Repair Algorithm 1 to generate feasible solutions.

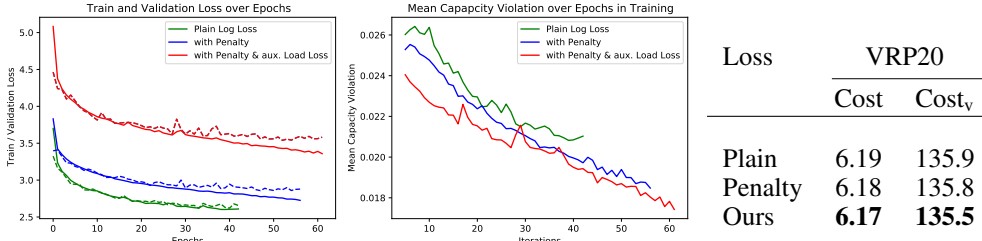

| Loss | VRP20 | |
|---|---|---|
| | Cost | Cost$_v$ |
| Plain | 6.19 | 135.9 |
| Penalty | 6.18 | 135.8 |
| Ours | **6.17** | **135.5** |

**Figure 3 & Table 4:** Ablation on the different Loss Structures Imposed on the Model.
The left sub-figure illustrates the train (solid line) and validation (dashed line) losses over epochs across the three loss configurations for the VRP with 20 customers; "Plain", i.e. no additional loss structures (green), "Penalty" (blue) and both "Penalty" and Load Loss structures (red). The right sub-figure displays the average capacity constraint violation of vehicles as training proceeds. Table

The loss functions on the left hand side of Figure 3 show that the losses including the auxiliary loss structures are overall higher, but can exhibit a higher delta between starting losses and final losses, which translates in better performance illustrated in Table 5.

## B.2   Soft-Assign Layer

In this section we want to demonstrate the empirical findings that a simple Softmax Normalization function is competitive to the SoftAssign Score Normalization Algorithm by Kaempfer & Wolf (2018).

**Table 5:** Comparison of Score Normalization Techniques

| Score Normalization | VRP20 | | | VRP50 | | |
|---|---|---|---|---|---|---|
| | Cost | $\text{Cost}_v$ | $t/$inst | Cost | $\text{Cost}_v$ | $t/$inst |
| SoftAssign | 6.23 | 135.5 | 0.057s | 10.85 | 346.3 | 0.241s |
| Softmax | 6.17 | 135.5 | **0.050**s | 10.77 | 346.1 | **0.170**s |

## B.3   Generalization

This section documents how the supervised VRP models trained on graph sizes of 20, 50 and 100 generalize to the respective other problem sizes, as well as to more realistic problem instances (Uchoa100) published in Kool et al. (2021) that are generated for graph-sizes of 100 following the data distribution documented in Uchoa et al. (2017). The underlying dataset in Table 6 is the rejection-sampled version of the dataset used in Kool et al. (2019).

**Table 6:** Generalization Ability of the Permutation Invariant VRP Model

| Training Distribution | VRP20 | | VRP50 | | VRP100 | | Uchoa100 | |
|---|---|---|---|---|---|---|---|---|
| | Cost | $\text{Cost}_v$ | Cost | $\text{Cost}_v$ | Cost | $\text{Cost}_v$ | Cost | $\text{Cost}_v$ |
| VRP20 | **6.16** | **135.5** | 11.06 | **344.8** | 17.33 | **858.2** | 19.68 | 904.1 |
| VRP50 | 6.28 | 136.3 | **10.76** | 346.1 | **16.85** | 860.2 | **19.41** | 897.5 |
| VRP100 | 6.31 | 136.0 | 11.04 | 345.5 | 16.93 | 859.6 | 19.42 | **896.6** |

Table 6 shows that in general and as expected, the models trained on a particular distribution perform best with regards to the objective cost, when tested on the same distribution.

## B.4   Target Solutions

In this section we provide additional head-to-head comparative results against the underlying target-generating method OR Tools (Perron & Furnon (2019)). The results are generated on the rejection sampled version of the Test-set used in Kool et al. (2019). We compare the OR-Tools method with our results that guarantee a solution to be found and set the search limit for OR-Tools Guided Local Search (GLS) to 10 seconds.

It is worth mentioning that with the indicated setting of the GLS, for 18, 7 and of the test instance, for the VRP20, VRP50 and VRP100 no solution could be found.

**Table 7:** Comparison to Target-generating Method OR-Tools on rejection-sampled datasets

| | VRP20 | | | VRP50 | | | VRP100 | | |
|---|---|---|---|---|---|---|---|---|---|
| | Cost | $\text{Cost}_v$ | $t/$inst | Cost | $\text{Cost}_v$ | $t/$inst | Cost | $\text{Cost}_v$ | $t/$inst |
| Ours | 6.17 | 135.5 | 0.05s | 10.77 | 346.1 | 0.16s | 16.93 | 859.6 | 0.84s |
| OR-Tools | 6.06 | 139.1 | 8.00s | 10.64 | 345.6 | 10.00s | 16.40 | 855.3 | 12.00s |

### B.5 Effectiveness of Post-processing

The Table 8 compares the effects of post-processing on both our model and the AM baseline model. The AM model is chosen as for this baseline, the resulting tours for each of the test instances can be easily retrieved and hence the postprocessing scheme can be run on top of those initially retrieved tours. However, we note that the additional post-processing procedure took an additional 3.5 minutes for the VRP20, 20 minutes for the VRP50 and 1 hour for the VRP100 in terms of runtime.

**Table 8:** Comparison of models with and without Postprocessing

| Method | VRP20 | | VRP50 | | VRP100 | |
|---|---|---|---|---|---|---|
| | Cost | $\text{Cost}_v$ | Cost | $\text{Cost}_v$ | Cost | $\text{Cost}_v$ |
| Without Postprocessing | | | | | | |
| Ours | 6.52 | 141.0 | 12.28 | 360.0 | 23.89 | 893.0 |
| AM (greedy) | 6.33 | 147.4 | 10.90 | 357.0 | 16.68 | 888.4 |
| With Postprocessing | | | | | | |
| Ours | 6.17 | 135.5 | 10.77 | 346.1 | 16.93 | 859.6 |
| AM (greedy) | 6.17 | 140.9 | 10.62 | 347.0 | 16.68 | 859.0 |

### B.6 Decoding Procedure Comparison to Kaempfer & Wolf (2018)

The most effective methodological contribution is the decoding process of solutions. In this section, we provide comparative results to the original decoding procedure in Kaempfer & Wolf (2018). Kaempfer & Wolf (2018) employ the "SoftAssign" normalization (mentioned in sub-section B.2) to obtain the probability distribution of tours and decode the discrete tours by a Backtracking or "Beam Search" procedure. In contrast to the Beam Search procedure we decode the solution greedily followed by a repair operation and a final postprocessing scheme.

Table 9 shows that the Beams Search decoding in Kaempfer & Wolf (2018), even though originally employed to solve the mTSP, is not capable of finding valid VRP solutions reliably. A simple greedy decoding followed by the proposed repair procedure increases the percentage of solved instances already by roughly 30%.

**Table 9:** Comparison of the original (mTSP) Decoding procedure in Kaempfer & Wolf (2018) (SoftAssign + Beam Search) and our algorithmic repaired greedy decoding (Repaired Greedy + Postprocessing) for the VRP with 20 nodes. "Beam Search 20" and "Beam Search 200" refer to the size of the beam, which is equal to 20 and 200 respectively. The test dataset evaluated on, is the rejection-sampled VRP20 dataset.

| Decoding Procedure | Cost | Percentage solved | Time |
|---|---|---|---|
| SoftAssign + Beam Search 20 | 7.42 | 42.34% | 11min |
| SoftAssign + Beam Search 200 | 7.25 | 69.00% | 15min |
| Repaired Greedy | 6.52 | **99%** | **8.0min** |
| Repaired Greedy + Postproc. | **6.17** | **99%** | 8.3min |

## C Methodological Details

### C.1 Entity Encoding

A VRP instance $X$, which defines a single graph theoretical problem, is characterised by the following *three entities*:

- A set of $N$ customers. Each customer $i$, $i \in \{1, ..., N\}$ has a demand $q_i$ to be satisfied and is represented by a feature vector $\mathbf{x}_i^{\text{cus}}$, which consists of its two-dimensional coordinates and the normalized demand $q_i/Q$.

- A collection of $M$ vehicles, each vehicle $k \in \{1, ..., M\}$ disposing a capacity $Q$ and characterised by the features $\mathbf{x}_k^{\text{veh}}$. The corresponding feature vector incorporates a local $(1/k)$ and global $(k)$ numbering, capacity $Q$ and the aggregated load that must be distributed.

- The depot node at the $0$-index of the $N$ customer vertices which is represented by a feature vector $\mathbf{x}^{\text{dep}}$ comprising the depot coordinates as well as a centrality measure indicating the relative positioning of the depot w.r.t. the customer locations.

## C.2   Vehicle Costs

Concerning the vehicle costs, Table 10 summarizes the different fixed costs, while accounting for the different vehicles capacities.

**Table 10:** Fixed Vehicle Costs

| Problem Size ($N$) | Vehicle | Capacity ($Q$) | Cost ($c_v$) |
|---|---|---|---|
| 20 | A | 30 | 35 |
| 50 | B | 40 | 50 |
| 100 | C | 50 | 80 |

In real-world use cases it is indeed justified that larger vehicles, disposing more capacity, also inquire more costs if we assume that not only buying or leasing these vehicles plays a role, but also the required space (e.g. adequate garage) or maintenance costs.

## C.3   Leave-one-out Pooling

The max-pooling operation on the one hand enforces the interaction of learnable information between entities and on the other hand mitigates self-pooling, i.e. each entity element's context vector consists only of other inter- and intra-group information, but not its own information. When the contexts to be pooled for element $r \in \{1, \ldots, l_g\}$ stem from its own entity, the leave-one-out pooling is performed in the Max Pooling Layer:

$$c_{g,r_g}^{(l)} = \text{pool}(h_{g,1}^{(l)}, \ldots, h_{g,r-1}^{(l)}, h_{g,r+1}^{(l)}, \ldots, h_{g,l_g}^{(l)}) \quad \text{for g} \in \{\text{dep, cus, veh}\} \tag{14}$$

When on the other hand, the contexts for element $r \in \{1, \ldots, l_g\}$ are pooled from another entity $g'$ that $r$ does not belong to, regular pooling is performed:

$$c_{g',r_g}^{(l)} = \text{pool}(h_{g',1}^{(l)}, \ldots, h_{g',l_{g'}}^{(l)}) \quad \text{for g} \in \{\text{dep, cus, veh}\} \tag{15}$$

## C.4   Pseudo-Greedy Decoding

The pseudo-greedy decoding described in section 4.2 transforms the doubly-stochastic probability tensor $\hat{\mathbf{Y}}$ into a binary assignment tensor $\hat{\mathbf{Y}}^*$, from which the predicted vehicle loads (required for the train loss) can be deduced. Algorithm 2 below shows the decoding procedure.

---

**Input:** $\hat{\mathbf{Y}} \in \mathbb{R}^{M \times N' \times N'}$
**Output:** $\hat{\mathbf{Y}}^* \in \{0,1\}^{M \times N' \times N'}$
1:   $\hat{\mathbf{Y}}^* \leftarrow \{0\}^{M \times N' \times N'}$ ;                          ▷ Initialize $\hat{\mathbf{Y}}^*$
2:   $\hat{\mathbf{Y}}^c \leftarrow \text{copy}(\hat{\mathbf{Y}})$ ;                  ▷ Create a copy of $\hat{\mathbf{Y}}$ for masking
3: **for** $k \in M$ **do**
4:     $i_{\text{curr}} \leftarrow \arg\max(\hat{\mathbf{Y}}_{k,0,:})$ ;        ▷ Assign "from-depot" customer for vehicle $k$
5:     $\hat{\mathbf{Y}}^*_{k,0,i_{\text{curr}}} \leftarrow 1.0$ ;             ▷ Fill $\hat{\mathbf{Y}}^*$ with the assigned start path
6:     $\hat{\mathbf{Y}}^c_{:,:,i_{\text{curr}}} \leftarrow -\infty$ ;            ▷ Mask assigned path for all vehicles
7:     **while** $i_{\text{curr}} \neq 0$ **do**
8:        $i_{\text{next}} \leftarrow \arg\max(\hat{\mathbf{Y}}^c_{k,i_{\text{curr}},:})$ ;       ▷ Select outgoing node from $i_{\text{curr}}$
9:        $\hat{\mathbf{Y}}^*_{k,i_{\text{curr}},i_{\text{next}}} \leftarrow 1.0$ ;           ▷ Fill $\hat{\mathbf{Y}}^*$ with the assigned path
10:       $\hat{\mathbf{Y}}^c_{:,:,i_{\text{curr}}} \leftarrow -\infty$ ;         ▷ Mask assigned path for all vehicles
11:       $\hat{\mathbf{Y}}^c_{:,:,0} \leftarrow \hat{\mathbf{Y}}_{:,:,0}$ ;     ▷ Reset "to depot" paths to original probabilities
12:       $i_{\text{curr}} \leftarrow i_{\text{next}}$ ;                           ▷ Reset index
13:     **end while**
14: **end for**
15: **return** $\hat{\mathbf{Y}}^*$

**Algorithm 2:** Pseudo-greedy Decoding

---

First the predicted binary assignment is initialised and a copy of the original probability tensor is created that can be used to mask the already visited customer nodes across vehicles and outgoing flows. After the starting customer of a vehicle $k$ is determined ($i_{\text{curr}}$) and stored in the output tensor $\hat{\mathbf{Y}}^*$, we loop through vehicle $k$'s path until the reset current vertex is again the depot, i.e. $i_{\text{curr}} = 0$. This is successively done for all vehicles and in batches of training instances, such that the algorithm proceeds reasonably fast.

## C.5   Train Loss

$$L_{k,p} = \min_{b_k \in \{0,1\}} -\left(\sum_{i=1}^{N}\sum_{j=1}^{N} \log \hat{\mathbf{Y}}_{k,i,j} \cdot \mathbf{Y}^{b_k}_{p,i,j}\right) + \alpha_{\text{over}} L_{\text{over}}(\text{load}(\hat{\mathbf{Y}}_{k,.,.})) \tag{16}$$

In a second step, after the optimal route direction has been determined, the vehicle permutation that renders the minimal loss is determined:

$$\mathcal{L} = \min_{\pi} \sum_{k=1}^{M} L_{k,\pi(k)} + \alpha_{\text{load}} |\text{load}(\hat{\mathbf{Y}}) - \text{load}(\mathbf{Y}_{\pi(k),.,.})| \tag{17}$$

## C.6   Delineation from the Permutation Invariant Pooling Network

There are five main components of the original Permutation Invariant Pooling Network that were extended considerably;

- The encodings are extentended; The customer entity is complemented by their demands, the depot encoding captures now the relative centrality of its location and the fleet encoding is enriched by the total demand to be distributed.

- Priming weights for the depot and customer groups in the pooling mechanism have been extended to incorporate the outer sum of demands in conjunction with the distance matrix to reflect that not only locations but also the demands of customers plays a role.

- Most changes were introduced in the decoding process. Instead of layering two simple 1D convolutions, we introduce an attention dot product to enhance the interaction of each vehicle and the edges to be allocated.

- The SoftAssign layers introduced by Kaempfer & Wolf (2018) seemed to be not as expressive as initially intended. Empirically we show in the experiments that even though the SoftAssign layers are technically effective, the same result can be achieved with a simple softmax normalization instead.

- Lastly we amended the loss for training the model in order to train the model also to learn *feasible* assignments of customers on to the tours.

# D    EXPERIMENT DETAILS

## D.1    HYPERPARAMETERS

The embedding size $d_m$, the hidden dimension of the shared fully-connected layer and the number of pooling layers $L$ is set to 256, 1024 and 8 respectively for the problem sizes of 20 and 50, due to GPU memory constraints we set $d_m = 128$ for the problem size of 100. The model is trained in roughly 60 epochs with a batch size of 128, 64 and 64 for the problem sizes 20, 50 and 100 respectively, using the Adam Optimizer (Kingma & Ba (2014)) with a learning rate of $10^{-4}$.

## D.2    NOTE ON BASELINE RE-EVALUATION

We want to note some particularities across the re-evaluations to accompany the findings in Table 2. First we note that the MDAM Model (Xin et al. (2021)) did not yield the same performances when evaluated in batches of thousand instances and in a single-instance based setting. This is due to the fact that the authors decided to evaluate their model in "training" mode during inference and consequently carried over batch-wise normalization effects[1]. Albeit, this might signal wanted transductive effects during inference, we found it more appropriate to document the results without transduction effects, as also the incorporation of these effects were not states in the publication. Concerning the NeuRewriter model (Chen & Tian (2019)) we only provide published results without the per-instance re-evaluation in Table 11, because even though we retrained the model according to the information in their supplementary materials, we were not able to reproduce the published results in a tractable amount of time. In order to re-evaluate the NLNS (Hottung & Tierney (2020)) and reproduce the published results, a GPU-backed machine with at least 10 processing cores was needed (as was indicated and confirmed by the authors). For the per-instance evaluations, we employed the batch-evaluation version of NLNS while setting the batch size to 2. The provided single instance evaluation method in Hottung & Tierney (2020) would require 191 seconds per instance, which again would be intractable to reproduce for the given test set of 10,000 instances. The L2I approach (Lu et al. (2020)) was not considered in this comparison as a single instance for the VRP100 requires 24 minutes (Tesla T4), which would lead to a total runtime of more than 160 days for the complete test set. Generally, the re-evaluation of the baseline models and the evaluation of our own approach were done on a Tesla P100 or V100 machine (except for the NLNS approach, since we required 10 processing cores - we re-evaluated this method on an A100 machine).

---

[1]The code is open source and the deliberate training mode setting is to be found in the file search.py: https://github.com/liangxinedu/MDAM/blob/master/search.py

### D.3 FULL RESULTS ON BENCHMARK DATASET

**Table 11:** Results for different RL and OR Models on 10000 VRP Test Instances Kool et al. (2019). Italic values constitute those that were published and also reproducible during our re-evaluation.

| Model | VRP20 Cost | $t$ | VRP50 Cost | $t$ | VRP100 Cost | $t$ |
|---|---|---|---|---|---|---|
| Gurobi | 6.10 | - | - | - | - | |
| LKH3 | 6.14 | 2h | 10.38 | 7h | 15.65 | 13h |
| | | | Batch Evaluation | | | |
| | Cost | $t$ | Cost | $t$ | Cost | $t$ |
| NeuRewriter | *6.16* | 22m | *10.51* | 35m | *16.10* | 66m |
| NLNS | *6.19* | 7m | *10.54* | 24m | *15.99* | 1h |
| AM (greedy) | *6.40* | 1s | *10.98* | 3s | *16.80* | 8s |
| AM (sampl.) | *6.25* | 8m | *10.62* | 29m | *16.20* | 2h |
| MDAM (greedy) | 6.28 | 7s | 10.88 | 33s | 16.89 | 1m |
| MDAM (bm30) | 6.15 | 3m | 10.54 | 9m | 16.26 | 31m |
| | | | Single Instance Evaluation | | | |
| | Cost | $t$/inst | Cost | $t$/inst | Cost | $t$/inst |
| NLNS (t-limit) | 6.24 | 2.41s | 10.96 | 2.41s | 17.72 | 2.02s |
| AM (greedy) | 6.40 | 0.04s | 10.98 | 0.09s | 16.80 | 0.17s |
| AM (sampl.) | 6.25 | 0.05s | 10.62 | 0.17s | 16.20 | 0.51s |
| MDAM (greedy) | 6.28 | 0.46s | 10.88 | 1.10s | 16.89 | 2.00s |
| MDAM (bm30) | 6.15 | 5.06s | 10.54 | 9.00s | 16.26 | 16.56s |
| Ours | 6.18 (94%*) | 0.05s | 10.81 (94%*) | 0.16s | 16.98 (98%*) | 0.82s |
| Ours** | 6.24 | 0.05s | 10.87 | 0.17s | 17.02 | 0.82s |

(*) Percentage of solved instances. (**) With the option of a guaranteed solution

### D.4 EVALUATION ON MORE REALISTIC TEST INSTANCES

In this section, we provide generalization results for the test set provided in Kool et al. (2021). We compare the LKH method and DPDP (Kool et al. (2021)) to the generalization results that were achieved after training our model on *uniformly*-distributed data of graph size 50 only.

Table 12 shows that, even though our approach is trained on uniformly distributed VRP50 data, it yields competitive results on the 10000 out of sample instances while being among the fastest methods.

**Table 12:** Comparison to DPDP (Kool et al. (2021)) and LKH for the VRP with 100 nodes on 10000 instances generated following the data generation process described in Uchoa et al. (2017). All values in the Table, except for "Ours", refer to the published results in Kool et al. (2021).

| Method | Cost | Time (1 GPU) |
|---|---|---|
| LKH | **18133** | 25H59M |
| DPDP 10K | 18415 | **2H34M** |
| DPDP 100K | 18253 | 5H58M |
| DPDP 1M | 18168 | 48H37M |
| Ours | 19412 | **2H38M** |

