# OpenReview forum: "Supervised Permutation Invariant Networks for solving the CVRP with bounded fleet size"
_ICLR.cc/2022/Conference — ICLR 2022 Submitted_

### Official Review · Reviewer_DMGQ · 2021-11-02

**Correctness:** 3
**Technical Novelty And Significance:** 3
**Empirical Novelty And Significance:** 2
**Recommendation:** 5
**Confidence:** 4

**Details Of Ethics Concerns:**

No.

**Main Review:**

Strengths:
1. This paper considers CVRPs with fleet size constraints and vehicle costs, which have not been studied by learning-based methods before. The proposed supervised learning model provides comparable results for standard CVRP and outperforms other learning-based methods accounting for fixed vehicle costs.

2. The model requires less training time when compared with other RL-based algorithms.

3. The model designs a special architecture to handle the permutation-invariant structure of CVRP, which is novel and insightful.

Weaknesses:
1. The whole framework is based on supervised learning, which requires a bunch of training data. For large-size CVRP, it is hard and time-consuming to get a near-optimal solution. The experiment results do not provide evidence that this model has any generalizability to large-scale problems, which could be a weakness of this paper.

2. The original definition of loss in equation (9) looks very computational-heavy, as it requires to compare every permutation, which the authors also mention it 'The calculation of the loss formulation in Equation 9 would induce considerable computational overhead, therefore a less memory-intensive formulation of the same loss is implemented (see Appendix C.5)'. However, it is unclear to me why the old loss function can be replaced with the new one. I cannot find any proof or explanation. I would expect the authors to provide more details.

Some comments:
1. For training time comparison, I think the authors should also include the time of generating training sets for your supervised learning model, as other RL-based algorithms do not need.

2. Some equations' are not very clear. For example, in equation (9), I cannot find the definition of $b_k$, I guess $b=b_k$. In equation (16), $b_k$ is defined, but I cannot find it in the equation.

**Summary Of The Paper:**

This paper proposes a supervised learning approach with a permutation invariant network to solve the Capacitated Vehicle Routing Problem (CVRP). Instead of assuming an unbounded fleet size, this model focuses on tackling the difficulty of constructing a complete tour plan with a specified fleet size and vehicle costs. The whole framework consists of an encoder, an information extractor, and a decoder. To guarantee the permutation invariance, the authors employ a separate embedding for each entity of the input with linear projection and design a permutation-invariant loss. The experiment results show that the proposed method works well for bounded fleet sizes CVRP and outperforms state-of-the-art RL models when considering fixed vehicle costs. Compared with other state-of-the-art algorithms, this supervised learning-based model is faster to train and provides competitive solutions with fewer vehicles.

**Summary Of The Review:**

My decision is weak reject.

The paper proposes a supervised learning algorithm to solve CVRPs with fleet size constraints and vehicle costs. The authors design a novel permutation invariant network to deal with the combinatorial structure. The experiment results show that the model has comparable performance for standard CVRP and outperforms other learning-based methods accounting for fixed vehicle costs.

However, my major concern is the generalization ability of this model. For large-scale CVRP (for example N > 500), simple google’s OR-Tools usually cannot provide near-optimal solutions, while LKH3 is very time-consuming. Therefore, generating a training set will be a big problem. I would change my mind if authors could come up with some ideas to improve or solve this problem.

---

> ### Author Response · Authors · 2021-11-15
> **Rebuttal - Addressing Reviewer's Concerns**
>
> Thank you for your invested time and insightful review of our paper. We address the weaknesses and concerns stated in your Review below:
>
> Addressing Weakness 1:
> ----------------------
> Regarding the concern of framing the "CVRP as a supervised ML Problem" we want to refer you to our response (1) given to Reviewer 3 (Reviewer e8jg) above.
> As for the concern with regards to the model's ability to generalize to large-scale problem sizes, we agree that it is an important capability,
> but have to note that it is currently a difficult task that limits the current routing approaches in the ML research community in general. Nevertheless, we have added preliminary experimental results regarding the ability of our model to generalize in Appendix B.3 of our current revision.
>
> Addressing Weakness 2:
> ----------------------
> The more "memory-efficient" loss formulation in Appendix C.5 stems from Kaempfer & Wolf (2018), thus to clarify the loss computation in detail we want to refer to the explanations in section 4.2.2 of Kaempfer & Wolf (2018). In fact, it is such that the loss in Appendix C.5 is still calculated for each permutation of vehicles. However, instead of calculating the optimal loss per permutation and per route-direction at the same time (Equation (9)), which would require O(2^M M! M) matrix multiplications, the formulation in Equation (16) "pre-calculates" the optimal losses for all 2^MM! possible targets and the predictions to determine the optimal route-direction, b_k, for each predicted route k in the first step. Then, in a second step, Equation (17) determines the permutation that renders the minimal pre-calculated loss, by enumerating all possible permutations.
> Even though this last step does not require any matrix multiplications (only O((M+1)!) simple operations), it still includes the iteration of all possible permutations, which of course is a bottleneck in terms of memory and a potential weakness, which we plan to investigate and address in future work, as has been stated in the conclusion. Also concerning Equation (16), we want to thank you for noting the mistake and clarify that b_k is missing as an exponent for Y_{p,i,j}.
>
> Improving on the generalization ability to larger instances:
> ------------------------------------------------------------
> As already mentioned above and shown in Joshi, Chaitanya K., et al. (2020), generalization to large-scale instances seems to be a task that the whole ML-based Routing community still needs to address and solve.
> One potential and pragmatic idea to tackle the generalization issue considering our proposed framework is to partition larger problems into smaller ones,
> where these partitions could also potentially be learned in a preliminary step.
> Generating separate target solutions for these smaller partitioned problems could solve the mentioned bottlenecks of OR-Tools and LKH3 for generating solutions. In a second step, our proposed model can be run for each of these sub-problems separately and could provide a potential future road to tackle large-scale routing problems.
>
> Reference:
> Joshi, Chaitanya K., et al. "Learning TSP requires rethinking generalization." arXiv preprint arXiv:2006.07054 (2020).

---

> > ### Comment · Reviewer_DMGQ · 2021-11-30
> > **Response**
> >
> > Thanks for your response.
> >
> > ''
> > As for the concern with regards to the model's ability to generalize to large-scale problem sizes, we agree that it is an important capability, but have to note that it is currently a difficult task that limits the current routing approaches in the ML research community in general.
> > ''
> > I think one main reason for using the learning-based algorithm to solve combinatorial optimization problems is taking advantage of its generalizability, as classic optimization solvers usually beat the learning-based algorithm in terms of objective value. We can see works like [1, 2] already have good generalizability.
> >
> > Meanwhile, how to generate high-quality training data for large-scale CVRP still seems unclear to me.
> >
> > [1] Xin, Liang, et al. "NeuroLKH: Combining Deep Learning Model with Lin-Kernighan-Helsgaun Heuristic for Solving the Traveling Salesman Problem." Thirty-Fifth Conference on Neural Information Processing Systems. 2021.
> > [2] Ma, Yining, et al. "Learning to Iteratively Solve Routing Problems with Dual-Aspect Collaborative Transformer." Advances in Neural Information Processing Systems 34 (2021).

---

### Official Review · Reviewer_e8jg · 2021-11-03

**Correctness:** 2
**Technical Novelty And Significance:** 2
**Empirical Novelty And Significance:** 3
**Recommendation:** 5
**Confidence:** 4

**Main Review:**

##########################################################################

 Pros:

-The goal of this paper, which is to develop an ML-based approach to the CVRP, is of significant practical interest and appears to be the first to do so.

-The claim of achieving competitive results to baselines that do not guarantee a fixed fleet size is supported by the experiments.

-The unique benefit of the approach, which is that it requires fewer vehicles albeit at slightly longer tour lengths, is validated by the results (particularly, in Table 1). However, it is unclear why Cost_v is not shown in Table 2.

##########################################################################

Cons:

-The main contribution appears to be a formulation of the CVRP as a supervised ML problem. I don’t think this is motivated very well. Supervised learning requires a dataset of solved instances of the problem. This becomes prohibitively expensive to acquire for non-trivial problem sizes. To my understanding, this is why RL approaches have garnered most of the attention from this community up to now. Perhaps the authors can provide a convincing argument in support of pursuing supervised ML approaches in the rebuttal.

-I understand that at a high level the technical contribution is a deep learning architecture that can output a tour plan for a fixed fleet size and given CVRP problem instance, but I don’t know exactly what the novel technical component is. From my current understanding, a previous architecture used for the mTSP problem is being repurposed with slight modification. But it is not clear what these modifications are exactly based on the current presentation of the method, which is a bit difficult to parse, and it is hard to therefore assess this contribution. Other aspects of this framework, including the solution decoding algorithm, inference solution post-processing, memory-efficient loss, and auxiliary losses, seem to be ad-hoc with unclear novelty.

-In Section 2, the related methods are described but they are not contrasted with the contributions made by this paper.

-The ability of the model to generalize to out-of-distribution problem instances of the same size, and to problem instances of larger sizes than was trained on, is not explored in the experiments. This is a crucial aspect to test since it is necessary for real-world application.

-Realistic test instances were proposed in Uchoa et al., and the models in this work should evaluate on them.

-It is not clear why DPDP from Kool et al. 2021 is not included as a baseline.

-The claim that RL methods are cumbersome to train is not well supported in general. Indeed, the AM baseline is an RL method that achieves both slightly better tour lengths while being comparably efficient as the proposed method (Table 2). (Granted, it cannot solve the problems with fixed fleet size.)

-Please provide confidence intervals or std. dev. across multiple runs for the main quantitative results (Tables 1 and 2).

References:

-Uchoa, Eduardo, et al. "New benchmark instances for the capacitated vehicle routing problem." European Journal of Operational Research 257.3 (2017): 845-858.

#########################################################################

Suggestions for improvement:

-Figure 2 could use a more detailed caption with each high-level module explained. The content under “Encoder”, “Information Extraction” and “Decoder” in the text should match the figure.

-Please define the sizes of vectors and shapes of matrices when introducing variables.

-Clarify along what dimensions that softmaxes are applied in Equation 8 and to transform the compatibility scores of vehicles and edges.

-Prefer providing a text-based description of the (pseudo-)greedy decoding algorithm (Section 4.2) and deferring the pseudocode to the appendix. Also, the “pseudo” aspect of the decoding isn’t explained.

-Based on the ablation study in Appendix B1 for the auxiliary losses, it doesn’t seem like there is a significant difference in Cost/Cost_v performance. Therefore, it could help to simplify the presentation to remove the auxiliary losses from the model.



**Summary Of The Paper:**

This paper is the first to address the capacitated vehicle routing problem with a hybrid machine learning (ML) and algorithmic solution. They demonstrate that their approach is able to find good tour lengths for fixed fleet sizes. Solving the VRP with fixed fleet sizes is well motivated by the limitations faced by service providers.


**Summary Of The Review:**

I currently am recommending a borderline rejection for this paper. I believe the work is important since it is the first to propose a deep learning solution for the VRP with fixed fleet size. However, I have concerns related to the underlying motivation for the proposed supervised deep learning approach, the validity of a few of the claims and contributions, and have made significant suggestions to strengthen the experiments section. Altogether, this makes me feel the paper is not yet ready for publication. I welcome a response from the authors, particularly pertaining to my concerns related to the motivation and contributions.

================================================================================================

Update after rebuttal: While some of the minor concerns I raised were addressed by the paper revisions, the major concerns about the use of SL, the claim of superiority of the proposed end-to-end SL approach compared to RL approaches (particularly with regards to the efficiency experiment shown in Table 3), and the lack of clarity with respect to the precise technical contributions were not addressed. Therefore, I am maintaining my initial score of 5.

---

> ### Author Response · Authors · 2021-11-15
> **Rebuttal - Addressing Reviewer's Concerns**
>
> Thank you for taking the time to review our paper thoroughly.
> We appreciate this high quality feedback and will consider it in our ongoing revision of the paper.
>
>
> (1) Addressing Concerns with respect to the CVRP as a supervised ML Problem:
> -----------------------------------------------------------------------
> We are aware of the potential downside that supervised learning approaches can have for the CVRP in terms of the target solution generation for larger problem sizes. Nevertheless, popular RL-based methods have also shown to reach their performance limits for larger problem sizes.
> Besides, there are also the following arguments that speak in favor of supervised learning approaches for solving combinatorial optimization problems:
>
> - Even though we need to account for the time that target instances need to be generated, we would still consider the training procedure itself to be faster and less cumbersome compared to RL procedures, especially because the target generation process can be delegated directly to an OR heuristic. Moreover, the target generation via OR heuristics (e.g. OR Tools) can be executed in an embarrassingly parallel way, such that with suitableinfrastructure, the generation process is very fast.
> -We would also argue that, since supervised learning models are not as sensitive to hyperparameter settings compared to RL methods, our method is more suitable for practitioners aiming to amend the permutation invariant VRP model to their own needs and purposes.
> -Lastly, we think that there's great evidence in the recent literature that also supervised learning approaches can produce state-of-the-art results for complex combinatorial optimization problems. For example in Nair et al. (2020), where the authors tackle the Mixed Integer Linear Program with 10^5 variables with supervised learning, while quantitatively and qualitatively outperforming non-commercial solvers on the particular tasks.
>
>
> (2) Addressing concerns with respect to the level of technical contribution:
> ------------------------------------------------------------------------
> The components of the original Permutation Invariant Pooling Model that were added or changed are described in Appendix C.6.
> The main adaption of the original model employed in Kaempfer and Wolf is the way in which the tour-plans are decoded. We agree that we could have pointed out these methodological contributions more clearly and will in this regard expand the ablation study section in Appendix B. In this sense we also want to delineate our contribution more effectively in section 2 from the related work in the final revision.
> Generally we want to note, that our methodological contribution also consists in merging the benefits of both research fields, deep learning from the ML side and algorithmic flexibility from the OR side, into a hybrid and holistic framework that empirically solves the CVRP for bounded fleet sizes.
>
>
> (3) Addressing concerns with regards to the evaluation:
> ---------------------------------------------------
>
> - We agree that the ability of the model to generalize is important and that we may have overlooked testing this ability, by favoring the head to head comparison of our model against the state-of-the-art RL methods in the field. We provide preliminary results on this issue in the current revision of the paper in Appendix B.3, which will be updated with additional results in the final revision at the end of the week when additional experiments are finished.
> - The train time comparisons in Table 3 demonstrates that the training procedure for an exemplary RL method takes longer compared to training our supervised approach. The complexity of training is negotiable of course, but we still think that in general RL approaches exhibit an elevated risk of unstable training and getting the model to converge in reasonable time or even at all sometimes takes a lot of expertise, development effort and hyperparameter tuning, which plays in favor of supervised learning approaches, especially for non-RL experts.
> - The original realistic test instances in Uchoa et al. (2017) have eventually not been considered, because the problem sizes (N>300) exceed the maximum of what can be currently efficiently handled by our approach. We nevertheless included the Uchoa-Distribution Test instances in Kool et al. (2021) in the Generalization section (Appendix B.3) of our current revision. We will also compare head-to-head on the published results of DPDP for this dataset in the final revision.
>
> Furthermore, we want to thank you for mentioning the useful "Suggestions for improvement" and we are happy to consider these in the final revision of the paper (which we will upload until the end of this week).
>
> Reference:
>
> Nair, Vinod, et al. "Solving mixed integer programs using neural networks." arXiv preprint arXiv:2012.13349 (2020).

---

> > ### Comment · Reviewer_e8jg · 2021-11-28
> > **Response to authors**
> >
> > Thanks for your response to my feedback and questions!
> >
> > ### On SL vs. RL and the extent of the methodological contributions
> >
> > > *Nevertheless, popular RL-based methods have also shown to reach their performance limits for larger problem sizes.*
> >
> > I don’t believe there is strong evidence to support the claim that RL methods have already reached a performance limit in terms of problem size.
> >
> > The current experiments do not convincingly demonstrate that RL methods are more sensitive to hyperparameters than SL methods. For this, we would suggest:
> > * Showing the main quantitative results with error bars over multiple random seeds to quantify the variance for fixed hyperparameters
> > * Including the results of the hyperparameter tuning process. Showing error bars over the range of attempted hyperparameters with fixed seed would help quantify sensitivity to a particular choice of hyperparameters
> > * Reporting the time taken to tune the hyperparameters for each method (SL and RL) to quantify whether the time needed to tune RL methods is indeed significantly greater
> >
> > In Nair et al., they are strategically using SL to tackle sub-problems within a complex algorithmic framework for solving large-scale MIPs. This seems to be a clearly distinct approach compared to this work where SL is used to generate a solution end-to-end and then an algorithmic module is used for solution post-processing. This distinction is important since the approach in Nair et al. appears to be more scalable (although perhaps less general).
> >
> > I believe it remains to be seen whether SL or RL is superior for end-to-end approaches. My suggestion would be to de-emphasize or remove claims of superiority of SL vs. RL and to rather focus on clarifying the specific methodological contributions in the main text.
> >
> > ### On evaluation
> >
> > Thanks for adding the generalization experiment; the performance appears to be quite promising.
> >
> > On the efficiency of SL vs. RL in Table 3, my main concern was with the justification for choosing NeuWriter as the exemplary RL method to compare against. It would seem that AM is the best RL method for this comparison based on the fact that it achieves similar Cost/Cost_v vs. t/inst as reported Table 1 and Table 2. This point was not clarified.
> >
> > Thanks for adding the realistic test instance results to the appendix.

---

### Official Review · Reviewer_hmYm · 2021-11-04

**Correctness:** 3
**Technical Novelty And Significance:** 2
**Empirical Novelty And Significance:** 3
**Recommendation:** 6
**Confidence:** 3

**Main Review:**

The paper is clearly written, and nicely introduces the reader to the bounded CVRP problem. It provides ample details inits annex to help the interested reader dig further in the work. Last but not least, it improves quality of results compared to previous solutions while providing similar search time.

My main criticism is with regards to the evaluation. Since the technical solution is derived from the work of Kaempfer and Wolf, I would have liked to see the authors compare their solution against that of Kaempfer and Wolf both on the mTSP problem that Kaempfer and Wolf used in their paper and on the CVRP problem tackled in this paper.
Furthermore, since the authors introduce several refinements to the permutation invariant pooling network architecture, I would have liked to see a complete ablation study to get a sense of how much refinement helped. The paper only provides evidence that using softmax instead of softassign helps in table 5. In particular, I am curious to see how much of the improvement they get comes from the post processing step as opposed from the learnt representation.
Additionally, it would have been useful to see the optimal solution in the evaluation table. The authors were able to generate optimal solutions to build their training set, they should have also generated these solutions for their test set. This would have given us a sense of the significance of their improvement in QoR which is currently hard to evaluate since the baseline they used were designed to tackle a slightly different problem.
The author also claims outstanding training time compared to NeuRewriter. I am wondering why they didn't compare their training times against these of the baselines they used in table 1. Furthermore, they should explicitly state how time consuming was the generation of their large training set.

**Summary Of The Paper:**

The paper demonstrate how to encode the CVRP with fixed fleet size into a set of input features that can be processed using a neural network derived from the permutation invariant pooling network architecture from Kaempfer and Wolf. They train the neural network using the Kaempfer and Wolf loss extended to penalize infeasible solutions. Lastly, unlike Kaempter and Wolf who use a beam search to convert the predictions made by the neural network into actual solutions, they use a greedy approach complemented by an OR solver to locally improve the solutions.

**Summary Of The Review:**

The paper propose incremental improvements to improve the work of Kaempfer and Wolf and adapt it to tackle the CVRP with bounded fleet size problem. The evaluation of this works needs improvement to trully demonstrate the value of each of these improvements.

---

> ### Author Response · Authors · 2021-11-15
> **Rebuttal - Addressing Reviewer's Concerns**
>
> Many thanks for your insightful comments and taking the time for reviewing the paper.
>
> Addressing Evaluation Concerns:
> --------------------------------
> With the current experimental evaluation, we wanted to focus on the bench-marking results of our model with respect to the current reproducible state of the art.
> However, we agree that:
>
> 1. Further ablations on the explicit methodological amendments listed in Appendix C.6 would have been insightful for the reader. While Appendix B is a start, we will extend this section with further ablations and possibly also comparisons to the original model used in Kaempfer and Wolf (2018).
>
> 2. The empirical effect of the Post-processing and the comparison to the Target-generating Method should have been discussed - we added according preliminary results in Appendix B.4 and B.5 of the current revision, which will be completed and extended in the following days with a final revision at the end of the week (experiments are still running).
>
> Concerning the comparison of other baselines' training times, we decided not to re-train these models (and hence not record training times), because the authors of the baselines already provided model checkpoints that we used for our evaluation. Nevertheless, we agree that the times to generate the training sets should be stated to ensure a fairer comparison. We added an according statement in the current revision.

---

> > ### Comment · Reviewer_hmYm · 2021-11-29
> > **Response to authors**
> >
> > Thank toy for taking the time to run additional experiments to address some of my concerns. I would have loved to see further ablations, but I understand that the short amount of time you had for the rebuttal limits what you could achieve. That said, the additional experiments you managed to complete seem to indicate that the performance of your solution is due is very large part to your decoding and post-processing steps, neither of which is based on machine learning. Have you considered submitting your work to an OR focused conference which could be a better fit for your contributions ?

---

### Official Review · Reviewer_gFqt · 2021-11-06

**Correctness:** 2
**Technical Novelty And Significance:** 2
**Empirical Novelty And Significance:** 3
**Recommendation:** 3
**Confidence:** 3

**Main Review:**

One strength of this paper is that it identifies the common limitation of existing approaches in the literature: that they do not actually solve CVRP with hard constraints on the fleet size. Authors provide good explanation of why this violation is problematic for practical applications. Authors also propose a cost function that is motivated by prior work, which would facilitate future research.

Also, a good range of reinforcement learning baselines are considered in this work. This helps readers to understand the practical utility of the proposed method against alternatives.

However, I am not sure the proposed method is an effective way to impose fleet size constraint. The proposed model isn't really guaranteed to produce feasible solutions. In fact, a heuristic repair procedure is applied to meet the constraint, and then another heuristic post-processing with OR tools is used to improve the quality of the solution. It isn't clear to me why these methods are not applicable to existing approaches. In that sense, authors' modeling contribution seems to be orthogonal to the problem of enforcing the fleet size constraint. Also, the experiment section does not clearly describe the contribution of these heuristic post-processing steps. Ablation studies on these components would be desirable. In addition, the post-processing step in Section 4.3 is not described in detail.

The main experiment in Section 5.1 doesn't include the expert policy (Google's OR Tools) as a baseline. I think it is necessary, because it helps practitioners to understand in which situations they should be using conventional optimizers like OR Tools vs. ML methods, including the proposed one. I also believe the comparison of training time is not really fair- it is quite straightforward that an imitation learning algorithm is faster and easier to train. However, authors do not discuss the fact that their imitation learning method requires the expert policy to train, which is a dependency RL methods don't require.

Imitation learning.

Effect of each component is unclear.

**Summary Of The Paper:**

Authors tackle the problem of solving a capacitated vehicle routing problem (CVRP) when the cost of introducing an additional vehicle is introduced. Authors propose to use imitation learning with behavior cloning, and uses Google OR Tools as the expert policy. Then, the proposed model extract features with multiple layers of attention and make conditionally independent predictions across the adjacency matrix. It is nontrivial to extract the feasible solution from the probabilistic predictions, so greedy decoding, repair procedure, and post-processing with OR tools is employed.

**Summary Of The Review:**

Authors have done a great job identifying a critical missing piece in the existing ML methods for CVRP. However, the proposed method relies on heavy heuristic engineering to enforce the constraint, and authors' modeling contributions seem orthogonal to the main problem (bounded fleet size) they are solving.

---

> ### Author Response · Authors · 2021-11-15
> **Rebuttal - Addressing Reviewer's Concerns**
>
> Thank you for taking the time to review our paper. Below we want to address
> and clarify some of your concerns.
>
> Concerns regarding the method's impact and the guaranteed solution:
> ---------------------------------------------------------------------------------------------
> We deliberately propose a hybrid ML and algorithmic solution for solving the CVRP, instead of relying entirely on ML. We believe that further research in this area should merge achievements in both, the OR research community and the the ML research community. Some works in the recent literature have pledged that this is a prominent way to solve routing problems efficiently in the future (see for example Bengio et al. (2021)).
>
> Regarding the guarantee of finding a solution, we want to mention two points:
>
> 1. At no point we claimed that our model is guaranteed to find a feasible solution for a fixed number of vehicles. This task, as mentioned in section 3 of our paper is already in itself NP-complete and is, to the best of our knowledge, not solved by any ML-based method so far.
> 2. Even though not guaranteed, we want to note that our holistic framework empirically and reliably solves the CVRP for bounded fleet sizes and does so better than the RL baselines. Furthermore, to provide maximum flexibility, we can even guarantee to find a solution to the routing problem, albeit at the cost of requiring more vehicles.
>
> Concerns regarding the evaluation:
> -----------------------------------------------
> We agree that the experimental study of our method lacks some possible ablations, due to focusing on the provided bench-marking experiments.
> We will add results concerning the effectiveness of our method's components as well as our target-generating method as a baseline in section 5 in the final revision of the paper. Concerning the current version of the revision, we provided results on the effectiveness of the post-processing scheme and a comparison against our target-generating method OR-Tools in Appendix B.5 and B.4 respectively. Additional ablations will be provided in the final revision at the end of the week, since the experiments need more time to run.
>
> Reference:
>
> Bengio, Y., Lodi, A., & Prouvost, A. (2021). Machine learning for combinatorial optimization: a methodological tour d’horizon. European Journal of Operational Research, 290(2), 405-421.

---

> > ### Comment · Reviewer_gFqt · 2021-11-29
> > **Responses**
> >
> > From B.5, it does seem like the proposed method is strongly dependent on the post-processing, and AM can benefit from it as well from the post-processing. This, together with the fact that the decoding process relies on additional repair procedure, I am not sure the benefit of introducing a new model is sufficiently demonstrated.

---

### Author Response · Authors · 2021-11-15
**Updates concerning the current revision of the paper.**

We have uploaded a revision of the paper that comprises the following changes and additions:

- We corrected the Cost_v results of our method (a minor mistake has been noticed in the calculation of the number of tours of a solution)
- We have added a statement concerning training time and dataset generation in section 5.3.
- We have added a "Generalization" section in Appendix B.3 that assesses the ability of our model to generalize to different (and notably higher) problem sizes, including different data distributions. This section still needs to be completed until the final revision.
- We have added comparative results of our model and our used target-generating method (OR-Tools) in section B.4. This section will also be completed during the course of the week.
- We have added a section on the effectiveness of the post-processing scheme, where we compare our results and the results of a baseline (AM) with and without post-processing.

---

### Author Response · Authors · 2021-11-22
**Updates concerning the final revision of the paper.**

We have uploaded a (final) revision of the paper that comprises the following additions:

- Ablations regarding the generalization ability of our model in Appendix B.3 and the comparison against the target-generating method (OR-Tools) in B.4 are completed.
- A further section in Appendix B (B.6) documents and empirically validates our contributed decoding procedure with respect to the original decoding and post-processing scheme in Kaempfer & Wolf (2018).
- We have added further comparative experiments in Appendix D.4 concerning the Test set in Kool et al. (2021), where we compare against the DPDP method (Kool et al. (2021)) and LKH. The test instances in these experiments are generated from the data distribution in Uchoa et al (2017). The results for our model in D.4 refer to out-of-sample generalization performance results. We also aim to train our model on these more realistic instances to provide a fair head-to-head in-sample performance comparison on this dataset.

---

### Decision · Program_Chairs · 2022-01-20

**Decision:**

Reject

**Comment:**

This paper formulates and solves a capacitated vehicle routing problem (CVRP) in the presence of costs for deploying additional vehicles: a mixture of supervised learning, algorithms, and OR techniques is used. In particular, a mix of greedy decoding, repairing of the solution, and post-processing with OR tools is used to extract a feasible solution from the probabilistic prediction.

The paper makes a good case that existing methods do not solve the CVRP with a hard constraint on the fleet size.  On the other hand, there is a strong dependence on heuristic improvements: e.g., a strongly dependence on the post-processing, and an additional repair procedure for the decoding process. The authors are encouraged to investigate how such improvements would work with existing approaches: i.e., how novel the new model’s contributions are.